# Secondary Mitochondrial Dysfunction as a Cause of Neurodegenerative Dysfunction in Lysosomal Storage Diseases and an Overview of Potential Therapies

**DOI:** 10.3390/ijms231810573

**Published:** 2022-09-12

**Authors:** Karolina M. Stepien, Neve Cufflin, Aimee Donald, Simon Jones, Heather Church, Iain P. Hargreaves

**Affiliations:** 1Adult Inherited Metabolic Disorders, Salford Royal NHS Foundation Trust, Salford M6 8HD, UK; 2School of Pharmacy and Biomolecular Sciences, Liverpool John Moores University, Liverpool L3 3AF, UK; 3Willink Biochemical Genetics, St Mary’s Hospital, Manchester Foundation Trust, Manchester M13 9WL, UK

**Keywords:** secondary mitochondrial dysfunction, lysosomal storage diseases, Gaucher disease, Niemann–Pick disease, type C, mucopolysaccharidosis, neurodegeneration

## Abstract

Mitochondrial dysfunction has been recognised a major contributory factor to the pathophysiology of a number of lysosomal storage disorders (LSDs). The cause of mitochondrial dysfunction in LSDs is as yet uncertain, but appears to be triggered by a number of different factors, although oxidative stress and impaired mitophagy appear to be common inhibitory mechanisms shared amongst this group of disorders, including Gaucher’s disease, Niemann–Pick disease, type C, and mucopolysaccharidosis. Many LSDs resulting from defects in lysosomal hydrolase activity show neurodegeneration, which remains challenging to treat. Currently available curative therapies are not sufficient to meet patients’ needs. In view of the documented evidence of mitochondrial dysfunction in the neurodegeneration of LSDs, along with the reciprocal interaction between the mitochondrion and the lysosome, novel therapeutic strategies that target the impairment in both of these organelles could be considered in the clinical management of the long-term neurodegenerative complications of these diseases. The purpose of this review is to outline the putative mechanisms that may be responsible for the reported mitochondrial dysfunction in LSDs and to discuss the new potential therapeutic developments.

## 1. Introduction

Although individually rare, lysosomal storage disorders (LSDs) as a whole constitute one of the most common groups of inherited metabolic diseases, with an estimated incidence of approximately 1:5000 live births [1].

LSDs are a group of rare heterogeneous inherited metabolic diseases that are characterised by the accumulation of undigested or partially digested macromolecules in the lysosomes. The accumulation or storage of macromolecules may begin during the period of early embryonic development, and the clinical presentation of these disorders can vary from an early and severe clinical phenotype to a late-onset mild disease [1,2]. This accumulation of undigested macromolecules can disrupt normal cellular functioning and gives rise to the clinical manifestations of LSDs. Only a few LSDs lack pathology in the central nervous system (CNS), and CNS involvement is relatively common in the majority of these disorders; the resulting symptoms are often the most debilitating, in view of the accompanying neurodegeneration, and can present in multiple brain regions, including the hippocampus, cerebellum, thalamus, and cortex [2]. Apart from the CNS, LSDs can affect the different organs of the body, including the heart, lungs, eyes, bones, kidneys, and skin [3]. If untreated, the life expectancy of patients with these disorders may be significantly reduced [3].

The LSDs encompass a group of over 70 different inherited metabolic disorders which are primarily characterized by lysosomal impairment as the result of defective lysosomal enzymes or membrane proteins [4] (Figure 1). Although the majority of LSDs result from acidic hydrolase deficiencies, a considerable number of these disorders can result from defects in either lysosomal membrane proteins or the other non-lysosomal proteins that are required for the optimal functioning of the lysosomal system. LSDs can be classified biochemically according to the particular type of storage material that has accumulated as the result of lysosomal impairment, e.g., as sphingolipidoses, mucopolysaccharidoses, mucolipidoses, or neuronal ceroid lipofuscinoses [5]. Although lysosomes are primarily responsible for the physiological degradation and recycling of cellular constituents, they also play a fundamental role in a number of other cellular processes, including secretion, signalling, energy metabolism, and plasma membrane repair [5].

### 1.1. Lysosome

Lysosomes are membrane-bound organelles with an acidic lumen (pH 4.5–5.5) that contain several types of hydrolytic enzymes which are responsible for the degradation of specific substrates within the cell. The primary functions of the lysosome can be divided into three main types: degradation, cell signalling, and secretion [6], (Figure 1). Lysosomes are involved in the degradation and recycling of extracellular material via the process of endocytosis and the degradation and recycling of intracellular material by the process of autophagy [7]. Extracellular material is generally conveyed to the lysosome by the process of endocytosis through specific endocytic mechanisms, based on the nature of the material. Autophagy, which is the catabolic pathway used by cells for the conserved degradation of cellular components, is used to convey intracellular materials to the lysosome [7]. As the result of autophagy, organelles and macromolecules are recycled via autophagosome-mediated transport and fusion with the lysosome. The breakdown products resulting from autophagy can then be used to generate new cellular components and/or energy in response to cellular nutritional requirements [7]. Lysosomes are also capable of undertaking Ca^2+^ regulated exocytosis, secreting their luminal contents into the extracellular space, as well as repairing damaged plasma membranes [8]. Lysosomal exocytosis may also have a role in modulating cellular clearance [8]. Additionally, lysosomes play a role in cellular signalling; they are capable of sensing nutrient availability and activating the lysosome-to-nucleus signalling pathway that mediates the starvation response, as well as regulates cellular energy metabolism [6]. The importance of optimal lysosomal function for normal cell physiology is highlighted by the fact that lysosomal dysfunction has been associated with the degenerative processes of a number of human diseases, along with the process of ageing [9].

**Figure 1 ijms-23-10573-f001:**
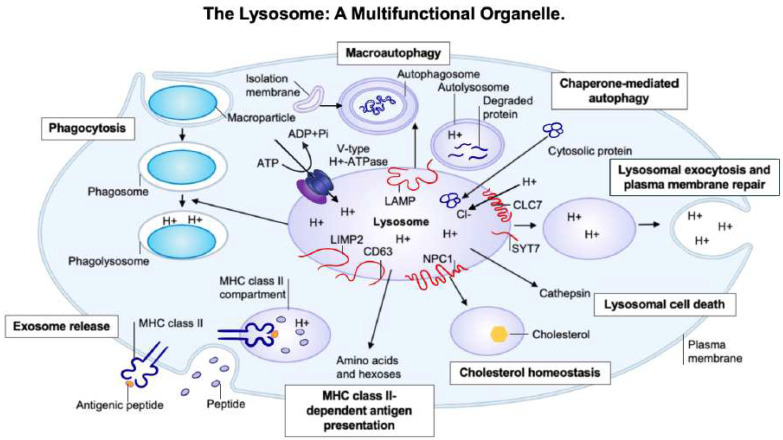
The lysosome: a multifunctional organelle.

### 1.2. Autophagy

Autophagy is the catabolic process through which cellular components, such as dysfunctional organelles, are first surrounded by vesicles known as autophagosomes and then recycled upon fusion of the autophagosome with the lysosome [6]. A blockage of autophagy, such as the impaired fusion of the autophagosome with the lysosome, can result in the accumulation of toxic material within the cell and may be an important contributory factor to the neurodegeneration in LSDs. An impairment in autophagy can initiate degenerative processes, not only in LSDs, but also in the more common age-related neurodegenerative diseases, including Parkinson’s (PD) and Alzheimer’s disease (AD) [10,11,12].

It has been reported that α-synuclein, a presynaptic protein which is prone to misfolding, can aggregate into protein structures, such as the Lewy bodies, and contribute to the pathogenesis of a number of neurodegenerative diseases as the result of an impairment in neuronal autophagy [11,12]. The lysosomal-dependent α-synuclein accumulation reported in LSDs may be a contributory factor to the neurodegeneration in these pathologies by inducing an α-synuclein chaperoning deficit and a consequent presynaptic failure [11,12].

### 1.3. Mitophagy

Mitophagy is a selective degradation of mitochondria often induced by mitochondrial respiratory chain (MRC) abnormalities, mitochondrial DNA damage, hypoxia, and dissipation of the membrane potential [2,12,13,14]. In this pro-survival mechanism, mitochondrial components are degraded and recycled. Damaged organelles become sequestered in the phagophores and subsequently, in the autophagosomes, instigating proteolytic degradation by a fusion with the lysosome. This mitophagy pathway moderates apoptotic signalling and mitigates cellular stress responses in a essential and unique quality control system [2,14,15,16,17,18]. The dysregulation of mitophagy may play a role in the pathophysiology of neurodegenerative diseases, as described below, and has been described in neuronal cells liberated from a mouse model of Gaucher disease [14].

### 1.4. Mitochondria

There is emerging evidence indicating the important role of mitochondrial dysfunction in the pathogenesis and progression of the neurodegeneration associated with LSDs [12,19]. The mitochondrial impairment in LSDs is caused by alterations in mitochondrial morphology, mass, and function [19]. The impaired mitochondrial metabolism in the CNS may result in the excessive production of mitochondrial reactive oxygen species (ROS), as well as a dysregulated calcium homeostasis. These metabolic disturbances can ultimately cause a mitochondria-induced apoptosis and associated neuronal degeneration [19].

Mitochondrial ATP production is requisite for neuronal metabolism and also serves as the major energy source for synaptic function [20]. The generation and maintenance of electrochemical gradients, vesicle release, and neurotransmitter synthesis strictly depend on the presence of functional neuronal mitochondria. This is illustrated by the fact that correct glutamatergic neurotransmission is ensured by perisynaptic astrocytes, which remove glutamate from the synaptic cleft and recycle it back to the pre-synaptic terminal via the glutamate–glutamine cycle, which is highly dependent on the availability of mitochondrial ATP, and even minor disruptions in mitochondrial metabolism can result in excitotoxicity and neuronal death. In children, LSDs represent a major cause of neurodegeneration [20] and is being increasingly linked to PD and other synucleinopathies in Gaucher disease, caused by homozygous and compound heterozygous mutations in the *GBA* gene. Interestingly, heterozygous mutations in the *GBA* gene were identified as the most frequent genetic risk factor for PD [21].

One of the primary functions of autophagy is to execute mitochondrial turnover [22]. Thus, it is not surprising that mitochondrial dysfunction is emerging as an important pathophysiological factor in LSDs. Considerable evidence supports the existence of crosstalk and reciprocal functional relationships between the mitochondrion and the lysosome [13,23]. While defective autophagy can affect mitochondrial quality-control pathways, it may also be a major contributor to the mitochondrial dysfunction associated with LSDs as the result of increased cellular ROS generation arising from the accumulation of damaged mitochondria [14]. This is discussed in more detail in Section 2 of the paper, along with its potential inhibitory effect on the mitochondrial function of macromolecules and protein aggregates, which accumulate as the result of impaired autophagy [22]. The mitochondrial proteins most susceptible to impairment in LSDs are the enzyme complexes of the mitochondrial respiratory chain (MRC) and ATP synthase [14]. Although the pattern of MRC deficiencies may vary amongst the different LSDs, it does not appear to be disease specific, and evidence of impaired mitochondrial function has been reported in animal models of the disease [24], as well as in patient fibroblasts [14]. This is discussed in more detail in Section 2 of the paper. Under healthy, non-disease conditions, there appears to be a reciprocal relationship between the mitochondrion and the lysosome. However, in LSDs, when the lysosomes become impaired, this impacts the function of the mitochondria, as previously discussed in the context of impaired autophagy and its consequences for mitochondrial function [14]. However, impaired MRC function will also impact on lysosomal function, since the acidity of the lysosome is predominantly maintained by the activity of a V-ATPase, which uses the free energy liberated from the hydrolysis of ATP to pump protons into the lumen of the organelle [14]. This is discussed in further detail in Section 2 of the paper. Disruption of mitochondrial function and homeostasis have been reported in several LSDs, including sphingolipidoses (Krabbe disease, Niemann–Pick disease, type C), some mucopolysaccharidoses, gangliosidoses, neuronal ceroid lipofuscinoses, and multiple sulfatase deficiency [14,22], and have been proposed possible mechanisms underlying neurodegeneration [15,19,24] (Table 1).

A common feature shared by mitochondria and lysosomes is that the impairment of their enzyme activities usually results in neurological pathologies, suggesting a potential functional connection between the mitochondria and the lysosomes [16,25,26]. Many LSDs resulting from defects in lysosomal hydrolase activity show neurodegeneration as the main clinical trait [4].

Mitochondrial dysfunction, including oxidative and nitrosative stress, is marked in a number of neurodegenerative diseases, and can contribute to the initiation and progression of the neurodegeneration [10,11,12]. Mitochondria can produce, and be modified by, ROS, and oxidative stress has also been implicated in the recruitment of specific mitochondrial autophagy proteins [27]. In addition, the accumulation of oxidatively-modified and aggregated proteins, along with mitochondrial impairment and the resultant deficit in energy-generating capacity, are prominent features in age-dependent neurodegenerative diseases [17]. Oxidative damage to proteins and organelles can contribute to the age-dependent accumulation of dysfunctional mitochondria and protein aggregates and has been associated with neurodegeneration. In addition, autophagy has been shown to play a protective role in the cellular response to ROS and reactive nitrogen species generation, as well as to toxic protein accumulation [18].

### 1.5. Peroxisome

Peroxisomes are ubiquitous single-membrane-bound organelles that have important roles in the degradation of very long chain fatty acids, the detoxification of ROS, and cell signalling [28]. Throughout the years, substantial evidence has been provided to indicate that peroxisomes and mitochondria exhibit a close functional interplay that can impact human health and development [29]. The connection between both organelles includes metabolic cooperation in the degradation of fatty acids, a redox-sensitive relationship, which was described in details elsewhere (review by Shrader et al. [29]). Furthermore, combined peroxisome-mitochondria disorders, with defects in organelle division, have been revealed. The role of peroxisomes in neurodegenerative disorders has been recognised previously [29].

Peroxisomes are responsible for the turnover of a number of pro-inflammatory mediators, such as arachidonic acid, and therefore, the impairment of this organelle may result in an inflammatory disease [30]. Neuroinflammation has been described in some LSDs, including Krabbe disease [31] and Gaucher disease [32].

### 1.6. LSDs and Neurodegeneration

CNS involvement is common in a number of LSDs, and neurodegeneration can occur in multiple brain regions, including the thalamus, hippocampus, cortex, and cerebellum [2] (Table 1). In LSDs, neuropathology produces unique temporal and spatial changes, which often involve early region-specific neurodegeneration and inflammation prior to the involvement of global brain regions. These changes occur because: specific storage metabolites exert differential effects on neuronal subtypes, varying proportions of macromolecules are synthesized in different neuronal populations, and there is differential neuronal vulnerability to the accumulated macromolecules. Activation of the innate immune system is also prevalent in the brain of LSD patients, which can directly contribute to the CNS pathology in these diseases [32]. Astrogliosis (activation of astrocytes) is another common feature of LSDs, impairing neuronal function as the result of inflammatory process known as glial scarring [32,33,34]. The additive detrimental effects that astrogliosis has on neuronal function is reiterated in animal models of LSDs [35,36]. The impairment of the intracellular transport pathways involving both endosomes and lysosomes is increasingly recognized as an underlying cause of neurodegenerative disease pathology in disorders such as AD and PD. Thus, the accumulation of damaged proteins and organelles (e.g., mitochondria) in neurons and glial cells has the capacity to overwhelm the intracellular recycling and degradative systems, thus exacerbating disease pathology. Endolysosomal ion channels have recently been reported to be important regulators of lysosomal exocytosis, pH, ion homeostasis, fusion and fission, endolysosomal trafficking, and autophagy [7].

Many of the LSDs clinically affect the nervous system, particularly the sphingolipidoses subgroup of these disorders, and almost all exhibit neuropathological changes. While mucopolysaccharidoses and mucolipidoses are LSDs primarily affecting the mesenchymal cells, they are also manifested clinically and pathologically in the CNS and peripheral nervous system (PNS) tissue, where the aberrant accumulation of the neuronal membrane components known as glycosphingolipids—especially ganglioside glycoproteins and glycosaminoglycans—can impair trans- and intercellular communication [4,34,37,38].

This review aims to link the way in which the aberration of the essential functions of the lysosome in LSDs may impact the mitochondria, causing an impairment of the organelle that could then contribute to disease pathophysiology in this group of conditions. It will also outline possible therapy strategies which, by targeting mitochondrial dysfunction, may be appropriate for the clinical management of patients.

## 2. Overview of Pathomechanisms of Mitochondrial Dysfunction in LSDs

Mitochondrial dysfunction is emerging as an important contributory factor to the neurodegeneration association with a number of LSDs, and although the causes of this neuronal dysfunction have yet to be fully elucidated, factors such as impaired mitophagy, the macroautophic degradation of the mitochondria, and oxidative stress are thought to be contributors [14]. However, impairment of the peroxisome as a secondary consequence of lysosomal dysfunction may also contribute to both the neurodegeneration and mitochondrial dysfunction associated with LSDs [28,39,40]. Lysosomes are thought to regulate lipid metabolism and cellular redox status through their action on peroxisome activity [41], and in a study by Tan et al. in 2019, the inhibition of lysosomal activity was found to result in a decrease in peroxisomal biogenesis, which was commensurate with a decrease in cellular catalase activity. Furthermore, an impairment in peroxisomal function has been reported in the mouse model of progressive neurodegenerative LSD, Niemann–Pick disease, type C (NPC), which is characterised by the accumulation of sphingomyelin and cholesterol in the lysosomes [42]. The peroxisomes form dynamic membrane contacts with lysosomes, which is required for intracellular cholesterol transport between the two organelles [43]. Therefore, a decrease in peroxisomal number or function as a consequence of lysosomal impairment may perturb intracellular cholesterol trafficking, leading to an accumulation of cholesterol within the lysosome and further exacerbating cholesterol homeostasis in NPC-patient neural cells. Furthermore, the increased mitochondrial membrane cholesterol content of cortical neurons, astrocytes, and isolated mitochondria derived from the brains of NPC mice has been associated with decreased ATP synthase activity and reduced cellular ATP levels [44]. The possibility that the impairment of oxidative phosphorylation was a consequence of increased mitochondrial membrane cholesterol content was further supported by the ability of the cholesterol chelator, methyl-cyclodextrin, to restore ATP synthesis to control cholesterol levels in NPC mouse brain mitochondria [44]. The increase in mitochondrial membrane cholesterol content is thought to adversely modify the physical properties of the membrane, which may then reduce the proton motive force and consequently, the membrane potential, resulting in a concomitant decrease in the ATP synthetic capacity of the organelle [44]. An altered mitochondrial membrane lipid composition was shown to be associated with impaired MRC function in PD, as the possible result of aberrant MRC super-complex formation [45]. The accumulation of mitochondrial membrane cholesterol was also reported to impair the transport of the tripeptide cellular antioxidant reduced glutathione (GSH) from its site of synthesis in the cytosol into the mitochondrion, as the transport of GSH is carrier-mediated and dependent on the inner membrane fluidity [46]. A deficit in mitochondrial GSH would be expected to leave the enzymes of MRC, the phospholipid milieu of the inner mitochondrial and mitochondrial DNA, vulnerable to ROS-induced oxidative damage [47].

An impairment in autophagy as a consequence of lysosomal dysfunction has been reported to result in an accumulation of functionally compromised peroxisomes, which exhibit an imbalance between the ROShydrogen peroxide generation, as a by product of their acyl-CoA oxidase activity, and their antioxidant catalase enzyme activity, which can eventually result in enhanced cellular oxidative stress, potentially impairing MRC function [40,47]. Moreover, peroxisomal biogenesis defects arising as the result of lysosomal dysfunction may cause an impairment in the synthesis of the phospholipid antioxidant species plasmalogen, which may reduce the capacity of the cell to detoxify ROS, again making the MRC more vulnerable to oxidative damage. Decreased plasmalogen levels have been reported in red blood cells from patients with Gaucher disease, with the level of this antioxidant being found to negatively correlate with the total disease burden. However, no evidence of peroxisomal dysfunction was found in the red blood cells [48].

The accumulation of damaged and dysfunctional mitochondria has also been reported in a number of different LSDs and has been associated with impaired mitophagy [2]. Furthermore, the accumulation of mitochondria with impaired MRC function may cause an increased generation of cellular ROS and potentially cause oxidative stress-induced impairment of MRC function. Interestingly, evidence of increased oxidative stress has been widely reported in patients with the LSDs, although the origin of this ROS generation has yet to be fully established in a number of these diseases, which may not be exclusively mitochondrial in origin [22]. In addition to ROS generation, MRC dysfunction in LSDs may also originate from the inhibition of the enzymes by the macromolecules and protein aggregates which accumulate in the cytosol, as a consequence of impaired autophagy [22]. The impairment of MRC enzyme activities by accumulated lysosomal storage material has been indicated in Fabry disease [49] and Mucopolysaccharidosis III (MPSIII; [23]). MRC dysfunction in LSDs may also result from a deficit in the level of the MRC electron carrier and cellular antioxidant coenzyme Q10 (CoQ10), which in MPSIII, is associated with the malabsorption of this compound from the gut, along with a diminution in circulatory pyridoxal phosphate (PLP)—the active form of vitamin B6—status, which is an essential cofactor in the CoQ10 biosynthetic pathway [50]. A recent study by Montero et al. [51] indicated that the deficit in CoQ10 status may not be restricted to patients with MPSIII, but may be present in all types of MPS, apart from Hurler–Scheie and Maroteaux–Lamy syndromes. It was speculated that the accumulation of heparin sulphate (and perhaps other mucopolysaccharides) in MPS may create adducts with PLP, leading to a loss of vitamin B6 and consequently, to diminished CoQ10 concentrations [51]. The diminution in CoQ10 status reported in NPC patients has been associated with increased oxidative stress, although it does not appear to be a consistent finding is all patients with this condition [52].

One of the most common of the LSDs is Gaucher disease, which is caused by mutations in the *GBA1* gene, resulting in impaired beta-glucocerebrosidase and the ensuing accumulation of the enzyme’s substrate glucocerebroside within the lysosome, along with compromised lysosomal activity [53] resulting in mitochondrial dysfunction and free radical damage [38]. Gaucher disease can be classified into three clinical subtypes, according to the age of onset and degree of neurological involvement. Patients without CNS involvement are categorized as type I, whilst those with CNS involvement are categorized as types II and III [54]. Interestingly, a number of multicenter genetic studies have indicated that mutations in the *GBA1* gene are a risk factor for PD [21,37,55].

In view of the association between PD and mitochondrial dysfunction [56], along with the link between PD and Gaucher disease, a number of studies have assessed the possible cause of MRC dysfunction in Gaucher disease, and an overview of these mechanisms is outlined in Figure 2.

A study in the mouse model of Gaucher disease type II has indicated evidence of impaired macroautophagy, as well as a defective ubiquitin-proteasome system involved in the degradation of damaged and misfolded proteins [54]. Impairment of these degradation pathways has been associated with the cerebral accumulation of dysfunctional mitochondria protein aggregates (such as the neural protein, α-synuclein), along with the compounds glucocerebroside and glucosylsphingosine, which can be cytotoxic at high levels [57,58]. Furthermore, α-synuclein has been reported to directly impair MRC complex I activity in vitro in isolated mouse liver mitochondria [59]. Accumulated damaged and fragmented mitochondria are normally removed from the cell by the process of mitophagy. However, in neurons and astrocytes from the mouse model of Gaucher disease, type II dysfunctional mitochondria were not found to be marked for turnover by the PINK 1-Parkin mitophagy pathway, and thus were found to accumulate within the cytosol of the neural cells [54]. In mitochondria with negligible membrane potential as the result of severely impaired MRC function, the mitochondrial membrane receptor PINK1 cannot be imported into the inner mitochondrial membrane, thus residing on the outer mitochondrial membrane, where it recruits the cytosolic E3-ubiquitin ligase Parkin, which then initiates mitophagy [54]. In contrast, however, the impairment of MRC function (decreased complex I and II-III activities) in neural cells from the mouse model of Gaucher disease II was found to result in an insufficient dissipation of mitochondrial membrane potential required for PARKIN recruitment [54]. This was possibly due to a reversal of MRC complex V activity, maintaining some degree of the membrane potential [54]. This phenomenon, along with other aberrations in the macroautophagy pathway, appeared to prevent the removal of dysfunctional mitochondria from the cell [54]. Finally, the accumulation of mitochondria with impaired MRC activity may result in a concomitant increase in ROS generation and therefore, a possible further loss in MRC function due to oxidative stressinduced impairment [47]. Furthermore, in common with other organellular membranes, the lysosomal membrane is subject to damage mediated by ROS, which may result in further impairment of the lysosome and the release of hydrolytic enzymes into the cytosol, leading to the proteolytic degradation of cellular systems and organelles, further exacerbating disease pathophysiology [60].

An important consequence of secondary mitochondrial dysfunction in LSDs is the potential loss of lysosomal acidification which may further impact the function of the organelle. The majority of lysosomal enzymes are acidic hydrolyses requiring an acidic environment of between pH 4.5–5.1 for optimal activity, which is predominantly maintained by a V-ATPase [61,62,63]. The V-ATPase uses the free energy liberated from the hydrolysis of ATP to pump protons across the lysosomal membrane into the lumen of the lysosome, and therefore, the acidification of the lysosome requires functioning mitochondria to generate the requisite ATP for the proton pumping [62]. Furthermore, a deficit in cellular CoQ10 status may also compromise the function of the lysosomal respiratory chain, which is also involved in maintaining lysosomal acidity [62].

## 3. Future Therapies

The current therapies available for the treatment of LSDs have significant limitations, specifically in their bio-distribution in target organs such as the brain, which is a critical issue, since many of these disorders are associated with CNS involvement. Neurodegenerative diseases in LSDs remain challenging to treat, and efficient or even curative therapies are not sufficient to meet patients’ needs. A combination of early diagnosis and different therapeutic approaches, such as enzyme replacement therapy (ERT) and small molecule, antibody, and gene therapies, etc., may be required. However, the concept of preventing lysosomal storage, increasing lysosomal degradation (autophagy), increasing lysosomal exocytosis, and enhancing lysosomal function appears to be a judicious approach [64].

In view of the emerging complications of LSD pathophysiology, novel therapeutic strategies are being considered based on an entirely different rationale, and these may represent potential adjunctive approaches to the treatment of LSDs (Table 2). These strategies are not directed towards the correction of the lysosomal enzyme defect and the causative gene mutations, or the modulation of the flux of substrates to the lysosomes, although they are targeted to the modulation of the pathways secondarily altered in LSDs [13]. Furthermore, the modulation of target proteins playing key roles in controlling lysosomal function, such as the TPCs (Na^+^-selective channels) and TRPML (Ca^2+^ channel) channels, may also be propitious targets for investigation. The modulation of TRPML/TPC may selectively ameliorate aberrant endolysosomal disease processes. Endolysosomal ligand-gated and voltage-gated ion channels appear to be potential therapeutic targets, comprising 13% of all the identified drug targets [7,65].

Aberrant activation inflammation is another potential therapeutic target, since neuro-inflammation has now been reported in several LSDs, and the pathways that are involved in the activation of the inflammasome are now being considered as additional potential therapeutic targets. Pentosan polysulfate, a mixture of semisynthetic sulfated polyanions, has been reported to have anti-inflammatory effects in some LSDs, particularly targeting the activation TLR4 and the resultant release of pro-inflammatory cytokines and the tumor necrosis factor (TNF)-α. This compound has been investigated in MPS type I and II patients, MPS animal models of type I, IIIA, and VI [66,67,68], and in in vitro models of Fabry and Gaucher disease [69]. Intraperitoneal high-dose aspirin reduced neuroinflammation in MPS type IIIB mice, with significantly reduced transcript levels of MIP-1a, IL-1b, and GFAP [70]. A therapeutic cocktail consisting of miglustat (a substrate-reducing agent), a non-steroidal anti-inflammatory drug to ameliorate neuroinflammation, and curcumin (Ca^2+^-modulator) was investigated in Niemann–Pick disease, type C1 mice and was found to maintain body weight and motor function, reduce microglial activation, and delay the onset of Purkinje cell loss [71].

Studies have also focused on targeting abnormalities in the autophagic pathway of LSD patient cells, and manipulation of this pathway has been proposed as a potential treatment option for Pompe disease, providing some promising results in animal models [72,73]. In vitro and in vivo overexpression of the transcription factor gene TFEB was found to enhance glycogen clearance and induce exocytosis, also resulting in some improvements in physical performance in the mouse model of Pompe disease [72,73]. Further experimental therapeutic strategies have focused on the ‘normalisation’ of intralysosomal calcium levels in Niemann–Pick disease, type C1 [74] and the reduction in oxidative stress in Krabbe disease [75]. Stimulation of the cytoprotective effect of the molecular chaperone HSP70 protein by the small-molecule drug arimoclomol in Niemann–Pick disease type C1 has been investigated in a clinical trial and has shown some evidence of clinical benefits [76].

The involvement of mitochondrial dysfunction in LSDs is increasingly being highlighted in the literature, and therapies aimed at reversing this impairment have been previously investigated in studies assessing the treatment of primary mitochondrial disease patients [19]. The pharmacologic reduction in mitochondrial ROS generation appears to be an appropriate target for these therapeutic strategies. Treatment of the midbrain neurons from the Gba−/− Gaucher disease mouse model with CoQ10 was found to reduce mitochondrial ROS production and restore mitochondrial membrane potential [54]. CoQ10, as well as other protein misfolding pharmacological chaperone therapies targeting mitochondria, have been shown to be promising in neuropathic Gaucher disease [77]. Similarly, antioxidant treatment with the GSH ethyl-ester in a in Niemann–Pick disease, type C1 mouse model restored mitochondrial function and prolonged survival [78]. Similarly, N-acetylcysteine (NAC) treatment prevented cell death in an oligodendrocytitic cell models of Krabbe disease [79,80]. Studies using the potent thiol antioxidant N-acetylcysteine (NAC) have reported evidence of neuroprotection in both in vitro and in vivo models of PD [81], as well as in human studies [82]; therefore, NAC may be an appropriate therapeutic candidate for studies in the treatment of LSDs. Indeed, a study by Fu et al. reported evidence of therapeutic benefit of NAC treatment in rodent models of Niemann–Pick disease, type C1 [83]. At present, there is a clinical trial running at the National Institute of Health Clinical Center in the USA to assess the therapeutic efficacy of NAC in the treatment of Niemann–Pick disease, type C patients (NCT03759639), although, no results from the investigation are available as yet. Furthermore, in view of the increase in cellular ROS generation that has been associated with an accumulation of dysfunctional mitochondria as the result of impaired mitophagy in some LSDs, the use of low dose uncouplers such as 2,4 dinitrophenol (DNP) may be a consideration [54,84]. Uncoupling decreases mitochondrial ROS generation through a number of mechanisms, including by decreasing the oxygen tension of the mitochondrion and therefore, the availability of oxygen for one electron reduction in the ROS, superoxide in the MRC [84].

Another potential therapeutic target in LSDs is the disturbed cellular calcium homeostasis reported in both animal models of the disease and in patients. Interestingly, in neurodegenerative diseases, lysosomal calcium signalling is emerging as a novel therapeutic target, and Raffaelo et al. [85] proposed the molecule nicotinic acid adenine dinucleotide phosphate (NAADP) as a potential activator of lysosomal calcium release [85]. By targeting the lysosomal two-pore channels (TPC) and TRPML (mucolipin family of transient receptor potential) channels, NAADP was found to induce local calcium release at the lysosome-endoplasmic reticulum (ER) contact sites and also to cause calcium-induced calcium release from the ER [85]. Furthermore, lysosomal calcium release was reported to be able to regulate autophagy through calcineurin-dependent dephosphorylation and the subsequent nuclear translocation of TFEB [86]. Therefore, these results suggest that the lysosome may function as a signalling hub for calcium homeostasis and autophagy. Several LSD studies support the involvement of the lysosome in disturbed calcium signalling and neurodegeneration. A study by Rigat et al. [87] investigated the use of diltiazem as a chaperone in Gaucher patient cells [87,88]. Depletion of lysosomal calcium stores in fibroblasts from NPC patients, along with reduced lysosomal calcium release, was reported to disrupt TFEB-mediated autophagy [74,89]. The transcription factor TFEB appears to be a promising therapeutic target in the treatment of certain LSDs. Overexpression of TFEB in neuronal stem cells from a mouse model of multiple sulfatase deficiency mice resulted in a profound reduction in glycosaminoglycans levels and the restoration of normal cellular morphology [90]. In addition, in a rat model of PD, activation of TFEB following rapamycin (mTOR inhibitor) treatment prevented α-synuclein induced neurodegeneration and further disease progression [91]. The protein kinase mTOR functions by phosphorylating TFEB at the lysosomal surface, blocking its translocation to the nucleus [92]. Additional therapeutic strategies to target downstream apoptosis pathways include the use of either the caspase-9 inhibitor taurine or the caspase-3 inhibitor Z-DEVD-FMK, and these have been reported to show some therapeutic efficacy in a neuronal cell model of Niemann–Pick disease, type C1 [93]. In addition, the use of the pan-caspase inhibitor Z-VAD-FMK in the astrocytes of a mouse model of G(M1)-gangliosidosis was found to modulate autophagy and decrease their sensitivity to oxidative stress [94]. Interestingly, the compound Z-DEVD-FMK could be successfully delivered across the blood–brain barrier of mouse brains using nanoparticle technology, therefore making this pan-caspase derivative more amenable for patient studies [95].

Another potential therapeutic avenue is the modulation of the signalling proteins sirtuins, which mediate a number of functions in neuronal physiology. SIRT1, the most notable of these proteins, is predominantly located in the nucleus, and it plays major roles in longevity and neuroprotection, which may reflect its ability to increase mitochondrial biogenesis through activation of the transcription factor PGC1α [96]. In a mouse model of infantile neuronal ceroid lipofuscinosis, treatment with the SIRT1-activator resveratrol was found to increase cellular ATP status and mitochondrial mRNA levels and to increase the lifespan of the mice [97]. Additionally, a role for sirtuins in mitophagy has been proposed.

Finally, abnormal GSK3-signalling has emerged as a potential pathophysiological factor in neurodegenerative diseases. GSK3 is a serine/threonine kinase that has been implicated in the development of familial and sporadic AD [98]. GSK3 inhibition was shown to promote mitochondrial transport along axons [99]. A number of GSK3-inhibitors have shown good bioavailability in the brain [100]. One of these compounds is L803-mts, which was found to restore mitochondrial trafficking along the axons in a NSC34 motor neuron-like cell model of Krabbe disease, in addition to restoring mitochondrial transport in sciatic nerves of the Twitcher mouse [19,101].

Initial experimental approaches that target mitochondrial pathways are showing some promising results in their ability to prevent mitochondria-induced apoptosis in neuronal models of LSD. These approaches, directed toward correction of the secondary abnormalities in LSDs, may help improve quality of life and slow disease progression. It has been proposed that the correction of these abnormalities may compliment existing therapies for LSDs. For example, in Pompe disease, the impairment of autophagy has been found to negatively impact the lysosomal trafficking of the recombinant enzyme used for ERT, reducing the therapeutic efficacy of this treatment [102]. Therefore, the possibility of enhancing autophagy in this condition may translate into improved lysosomal delivery of the therapeutic enzyme.

In all probability, other therapeutic targets will be identified in the future as the result of the precise characterization of the pathogenetic pathway of LSDs, opening new avenues for the treatment of LSDs.

**Table 1 ijms-23-10573-t001:** Mechanisms of neurodegenerative dysfunction in LSDs.

Lysosomal Storage Disorder	Clinical Manifestation of Neurodegenerative Dysfunction in Humans	Pathomechanism of CNS Involvement	Reference
Mucopolysaccharidosis		Neuroinflammation	[103,104,105]
I	Mental retardation (variable); skeletal, heart, respiratory, corneal abnormalities	Oxidative stress (animal models and human blood)	[106,107]
II	As above	Oxidative stress (human blood)	[108]
III A	Mental retardation, behavioural disturbances, hyperactivity	Reduced excitatory synaptic strength on the somatosensory cortex (mouse model); inhibition of soluble NSF attachment receptor (SNARE) complex assembly and synaptic vesicle recycling, possibly caused by perikaryal accumulation of insoluble α-synuclein and increased proteasomal degradation of cysteine string protein α, resulting in low availability of these proteins at the synaptic terminal	[23] [51]
III B	Mental retardation, behavioural disturbances	Oxidative stress (animal models); Golgi involvement	[109,110]
III C	Mental retardation, behavioural disturbances, hyperactivity	Primary accumulation of HS in microglial cells and neurons causes impaired autophagy, secondary neuronal storage of GM2/GM3 gangliosides and misfolded proteins, neuroinflammation, and abnormalities in mitochondrial energy metabolism, eventually leading to neuronal death; synaptic vesicles are reduced in the terminals of MPSIIIC hippocampal neurons; MPSIIIC neurons show alterations in distribution of excitatory synaptic markers and in transmission; progressive deficiency of mitochondrial function; Selective reduction in OXPHOS complexes Decreased coenzyme Q10	[111] [24] [111] [24]
Multiple sulphatase deficiency	Neurological deterioration, ichthyosis, skeletal Anomalies, and organomegaly	Autophagy and mitophagy Accumulation Fragmentation Decreased ATP content	[112] [113]
Niemann–Pick			
A/B	Severe deterioration of the central nervous system (CNS), along with visceral and cerebral sphingomyelin storage	TRPML1-mediated Ca^2+^-release is compromised	[114] [115]
C	Sub-acute nervous system involvement, with a comparatively moderate course and a less pronounced degree of visceral storage pathology; cerebellar ataxia, dysarthria, dysphagia, progressive dementia, and occasionally seizures; progressive neurological regression, seizures, spasticity	Synaptic pathology (mouse model); SNARE impairment; mitochondrial cholesterol accumulation; the concomitant primary storage of cholesterol and secondary storage of sphingomyelin appears to be a prime driver for NPC1 pathology, interfering with TRPML1 and TRPML1-dependent maintenance of lysosomal homeostasis Mislocalisation of sphingolipids from the Golgi apparatus to lysosomes, as evidenced by impaired trafficking of lactosylceramide in NPC1 cells Abnormal accumulation of lipids in NPC1 patient lysosomes resulting in secondary lysosomal storage by blocking TRPML1- and Ca^2+^-dependent lysosomal trafficking; this storage could be reverted by the TRPML agonist ML-SA1 Sphingosine storage that induces calcium depletion in lysosomes, possibly through an inhibitory effect on Na^+^/Ca^2+^ exchangers Neuroinflammation Peroxisomal dysfunction Decreased oxidative respiration/reduced ATP levels Increased susceptibility to oxidative stress Decrease in mitochondrial GSH Cytochrome c release	[35] [116] [117] [117] [118] [117] [74] [119] [120,121] [42] [122]
Gaucher disease I-III	Progressive neurological regression, seizures, spasticity (mainly III)	Neuroinflammation Dysregulated calcium homeostasis Decreased mitochondrial membrane potential Selective reduction of OXPHOS complexes Accumulation of APP and α-synuclein Reduced O_2_ consumption/reduced ATP levels	[119] [120,121] [115] [54] [123] [57]
Krabbe disease		Synaptic pathology (mouse model) Peroxisomal dysfunction—downregulates the peroxisome proliferator–activated receptor-alpha (PPAR-alpha). Decreased mitochondrial membrane potential Elevated ROS/reduced GSH Dysregulation of Ca^2+^ signalling Cytochrome c release	[124] [31,125] [80]
GM1 gangliosidosis	Progressive neurological regression, seizures, spasticity	Synaptic pathology (feline model) Neuroinflammation; enhanced autophagy and mitochondrial dysfunction Dysregulated calcium signalling Decreased mitochondrial membrane potential Increased vulnerability to oxidative stress Cytochrome c release	[126,127] [128] [129] [94]
GM2 gangliosidosis (Tay Sachs)	Progressive neurological regression, seizures, spasticity	Dysregulated ER calcium homeostasis	[115]
Mucolipidosis IV	Mental impairment, speech impairment, spasticity, neuroaxonal dystrophy, blindness	Lack of the endolysosomal ion channel mucolipin1/ TRPML1/MCOLN1, with evidence of lysosomal accumulation of gangliosides and heavy metals such as zinc and iron; Ca^2+^ abnormalities	[130] [14,131]
Fabry disease	Progressive motor and nonmotor neurodegeneration; it remains unclear whether these are associated with Parkinsonian neurodegeneration	Disrupt the autophagy-lysosomal pathway, leading to autophagosomal accumulation of phosphorylated a-synuclein in the mouse brain TRPML1-mediated Ca^2+^-release is compromised	[14,132] [114]
Neuronal ceroid lipofuscinoses	Dementia, motor disturbances, epilepsy, loss of vision, and early death	Neuroinflammation Mitochondrial dysfunction Reduced ATP levels Deficient mitochondrial Ca^2+^ buffering Mitochondrial vacuolation	[133] [134] [135]
Pompe disease	Limb–girdle muscle weakness	Mitochondrial calcium excess, increased reactive oxygen species, decreased mitochondrial membrane potential, and decreased oxygen consumption and ATP production	[14,136]

ATP—adenosine triphosphate; GM2/GM3—gangliosides; GSH—glutathione; HS—heparan sulphate; OXPHOS—oxidative phosphorylation; SNARE—soluble NSF attachment receptor.

**Table 2 ijms-23-10573-t002:** Treatment target and therapies trialed in vivo or in vitro.

Treatment Target	Potential Therapy	Lysosomal Storage Diseases in which the Therapy was Trialled
Abberant activation inflammation	Pentosan polysulfate	MPS I, II In vitro in MPS I, IIIA, VI
	Intraperitoneal high dose aspirin	MPS IIIB
	Miglustat	Niemann–Pick disease, type C1
Increase cellular energy	Resveratrol	Neuronal ceroid lipofuscinosis
Autophagic pathway	-	In vitro in Pompe disease
	Arimocromol	Niemann–Pick disease, type C1
	-	Krabbe disease
Reduction in mitochondrial ROS generation	Coenzyme Q10	Gaucher disease + Fabry disease
	GSH ethyl-ester	Niemann–Pick disease, type C1
	N-acetylcysteine	Krabbe disease Niemann–Pick disease, type C1
Lysosomal calcium signalling	Curcumin	Niemann–Pick disease, type C1
	Diltiazem	Gaucher disease
mTOR signalling/transcription factor TFEB	Rapamycin	Multiple Sulfatase Deficiency
Apoptotic pathway	Z-DEVD-FMK	Niemann–Pick disease, type C1
	Z-VAD-FMK	GM1 gangliosidosis
Modulation of sitrulins	Resveratrol	Infantile NCL
Inhibition of GSK-3-signalling	L803-mts	Krabbe disease

## 4. Conclusions

The pathophysiological factors, including neurodegeneration, neuroinflammation, impaired autophagy, and microgliosis, are generally considered to be the major causes of behavioural disturbance, developmental delay, and progressive neurodegenerative decline in patients with neurological LSD.

Our current knowledge of lysosome biology and function is still developing, and there is a need for their further elucidation that may then provide critical information on the pathophysiology of LSDs. The understanding of disease pathophysiology is paramount, as it has the potential to help identify novel therapeutic targets and to indicate new strategies approaches for the treatment of LSDs. The use of appropriate markers to indicate evidence of cerebral mitochondrial dysfunction and oxidative stress in LSDs, as well as to monitor therapeutic intervention, will also have an important bearing on the elucidation of disease pathophysiology and management. A combination of therapies is likely to be required to achieve optimal disease control in the absence of a single curative intervention.

## Figures and Tables

**Figure 2 ijms-23-10573-f002:**
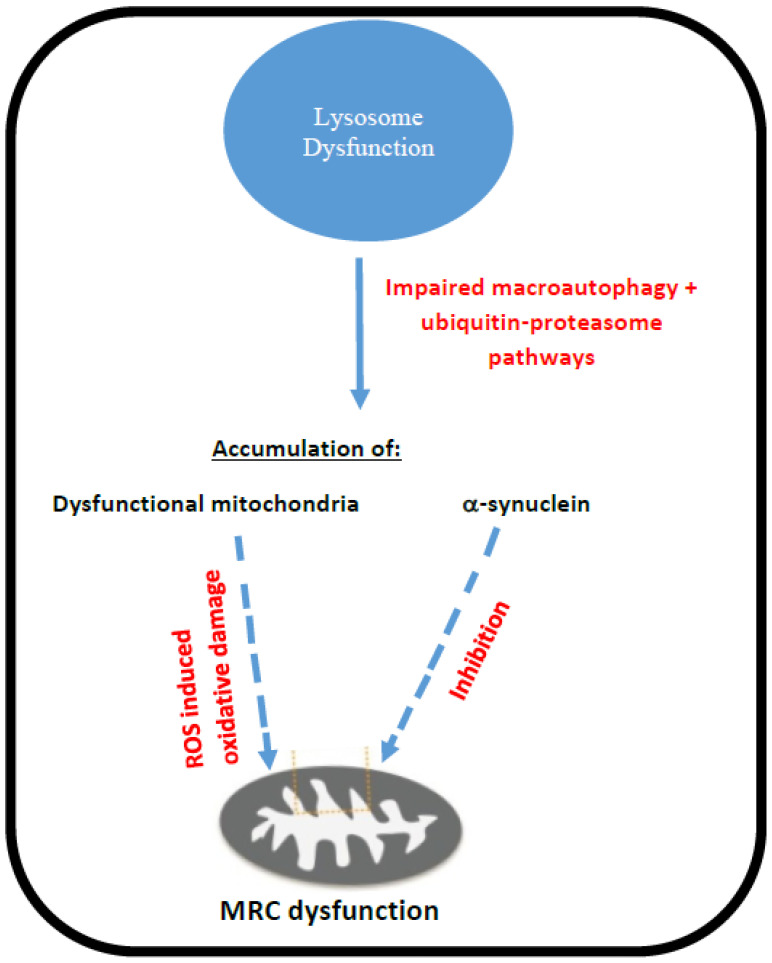
Putative mechanisms of MRC dysfunction in Gaucher disease.

## Data Availability

This is a review article and all the information presented is available in the cited references.

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
