# Peer review of "Secondary Mitochondrial Dysfunction as a Cause of Neurodegenerative Dysfunction in Lysosomal Storage Diseases and an Overview of Potential Therapies"

_ijms, 2022, doi:10.3390/ijms231810573_

Round 1
Reviewer 1 Report
Authors in the article "Secondary mitochondrial dysfunction as a cause of neurodegenerative dysfunction in lysosomal storage diseases and an overview of potential therapies" described the putative mechanisms that may be responsible for the reported mitochondrial dysfunction in LSDs and to discuss the new potential therapeutic developments.
Comments:
1. Paragraph 1.3 - can authors describe more deeply which mitochondrial proteins/protein complexes are damaged in LSDs and how in the case of lysosomal diseases the signal is transmitted from lysosomes to mitochondria. What is the relationship between - lysosomes and mitochondria in the case of LSDs
Author Response
- Can authors describe more deeply which mitochondrial proteins/protein complexes are damaged in LSDs?
RESPONSE: We thank the review for this comment and have added the following sentences in the manuscript:
` The mitochondrial proteins most susceptible to impairment in LSDs are the enzyme complexes of the mitochondrial respiratory chain (MRC) and ATP synthase [19]. Although the pattern of MRC enzyme deficiencies may vary amongst the different LSDs it doesn`t appear to be disease specific and evidence of impaired mitochondrial function have been reported in animal models of the disease [20] as well as in patient fibroblasts [19]. This is discussed in more detail in section 2 of the paper.`
- How in the case of lysosomal diseases the signal is transmitted from lysosomes to mitochondria ?
RESPONSE: We thank the review for this comment and have added the following sentences in the manuscript :
`While defective autophagy can affect mitochondrial quality control pathways it may also be a major contributor to the mitochondrial dysfunction associated with LSDs as the result of increased cellular ROS generation arising from the accumulation of damaged mitochondria [19]. This is discussed in more detail in section 2 of the paper together with potential
inhibitory effect on mitochondrial function of macromolecules and protein aggregates which accumulate as the result of impaired autophagy [16]. `
- What is the relationship between - lysosomes and mitochondria in the case of LSDs?
RESPONSE: We thank the review for this comment and have added the following sentences in the manuscript:
`Under healthy, non disease conditions there appears to be a reciprocal relationship between the mitochondrion and the lysosome. However, in LSDs when the lysosomes become impaired this impacts on the function of the mitochondria as previously discussed in the context of impaired autophagy and its consequences for mitochondrial function [19]. However, impaired MRC function will also impact on lysosomal function since the acidity of the lysosome is predominantly maintained by the activity of a V-ATPase which uses the free energy liberated from the hydrolysis of ATP to pump protons into the lumen of the organelle [19]. This is discussed in further detail in section 2 of the paper.
Reviewer 2 Report
Review titled ‘Secondary mitochondrial dysfunction as a cause of neurodegenerative dysfunction in lysosomal storage diseases and an overview of potential therapies’ written by Karolina Maria Stepien et al. outlines the putative mechanisms that may be responsible for the reported mitochondrial dysfunction in LSDs and further to discuss the new potential therapeutic developments. The manuscript is concise and well written.
Here are a few suggestions that are worth considering when you revise your manuscript.
1) Since mitophagy has been mentioned in 2. OVERVIEW OF PATHOMECHANISMS OF MITOCHONDRIAL DYSFUNCTION IN LSDs, it would be good to add mitophagy in the introduction part.
2) In the part of FUTURE THERAPIES, the authors first discuss the current therapies. It would be good to add content: is there any compound/drug available in clinic? Or under clinic trial?
The authors then discuss alternative options for treating LSD. The target mentioned above seem to belong to different pathways and it is difficult to see if there is any connection between them. It would be nice to have a diagram illustrating the possible targets mentioned in the manuscript (pathways or organelles), or at least a table summarizing all discussed targets for better readability.
Author Response
Review titled ‘Secondary mitochondrial dysfunction as a cause of neurodegenerative dysfunction in lysosomal storage diseases and an overview of potential therapies’ written by Karolina Maria Stepien et al. outlines the putative mechanisms that may be responsible for the reported mitochondrial dysfunction in LSDs and further to discuss the new potential therapeutic developments. The manuscript is concise and well written.
RESPONSE: Thank you
Here are a few suggestions that are worth considering when you revise your manuscript.
- Since mitophagy has been mentioned in 2. OVERVIEW OF PATHOMECHANISMS OF MITOCHONDRIAL DYSFUNCTION IN LSDs, it would be good to add mitophagy in the introduction part.
RESPONSE: A paragraph has been added:
Mitophagy is a selective degradation of mitochondria often induced by mitochondrial respiratory chain (MRC) abnormalities, mitochondrial DNA damage, hypoxia and dissipation of the membrane potential [2, 12, 19]. In this pro-survival mechanism, mitochondrial components are degraded and recycled. Damaged organelles become sequestered in phagophores and subsequently in autophagosomes that by a fusion with the lysosome instigate proteolytic degradation. This mitophagy pathway moderates apoptotic signalling and mitigates cellular stress responses in a essential and unique quality control [2, 19, 22, 27]. The dysregulation of mitophagy may play a role in the pathophysiology of neurodegenerative diseases as described below and has been described in neuronal cells liberated from a mouse model of Gaucher disease [19].
- In the part of FUTURE THERAPIES, the authors first discuss the current therapies. It would be good to add content: is there any compound/drug available in clinic? Or under clinic trial?
RESPONSE: Thank you for the suggestion. To the best of our knowledge there is no compound that we could be used in clinics and recommended by guidelines. In some individual cases with Fabry disease, co-enzyme Q10 was tried, but it was short-term and empirical rather than based on strong clinical evidence. At present there is a clinical trial assessing the therapeutic efficacy of N-acetyl-cysteine (NAC) in the treatment of Niemann-Pick disease C and we have included this information in the `Treatment` section:
`At present there is a clinical trial running at National Institute of Health Clinical Centre in the USA to assess the therapeutic efficacy of NAC in the treatment of Niemann-Pick disease type C patients (NCT03759639), although, no results from the investigation are available as yet.`
The authors then discuss alternative options for treating LSD. The target mentioned above seem to belong to different pathways and it is difficult to see if there is any connection between them. It would be nice to have a diagram illustrating the possible targets mentioned in the manuscript (pathways or organelles), or at least a table summarizing all discussed targets for better readability.
RESPONSE: Thank you for the suggestion and we have now incorporated the following table in the manuscript:
Table 2. Treatment target and therapies trialled in vivo or in vitro
|
Treatment target |
Potential therapy |
Lysosomal Storage Diseases the therapy was trialled |
|
Abberant activation inflammation |
Pentosan polysulfate |
MPS I, II In vitro in MPS I, IIIA, VI |
|
Intraperitoneal high dose aspirin |
MPS IIIB |
|
|
Miglustat |
Niemann-Pick disease type C1 |
|
|
Increase cellular energy |
Resveratrol |
Neuronal ceroid lipofuscinosis |
|
Autophagic pathway |
- |
In vitro in Pompe disease |
|
Arimocromol |
Niemann-Pick disease type C1 |
|
|
- |
Krabbe disease |
|
|
Reduction of mitochondrial ROS generation |
Coenzyme Q10 |
Gaucher disease + Fabry disease |
|
GSH ethyl-ester |
Niemann-Pick disease type C1 |
|
|
N-acetylcysteine |
Krabbe disease Niemann-Pick disease type C1 |
|
|
Lysosomal calcium signalling |
Curcumin |
Niemann-Pick disease type C1 |
|
Diltiazem |
Gaucher disease |
|
|
mTOR signalling/ transcription factor TFEB |
Rapamycin |
Multiple Sulfatase Deficiency |
|
Apoptotic pathway |
Z-DEVD-FMK |
Niemann-Pick disease type C1 |
|
Z-VAD-FMK |
GM1 gangliosidosis |
|
|
Modulation of sitrulins |
Resveratrol |
Infantile NCL |
|
Inhibition of GSK-3-signalling |
L803-mts |
Krabbe disease |